# A New Technological Advancement of the Drug-Induced Sleep Endoscopy (DISE) Procedure: The “All in One Glance” Strategy

**DOI:** 10.3390/ijerph17124261

**Published:** 2020-06-15

**Authors:** Michele Arigliani, Domenico M. Toraldo, Filippo Montevecchi, Luana Conte, Lorenzo Galasso, Filippo De Rosa, Caterina Lattante, Enrico Ciavolino, Caterina Arigliani, Antonio Palumbo, Michele De Benedetto, Claudio Vicini

**Affiliations:** 1ENT Unit, “V.Fazzi” Hospital, 73100 ASL Lecce, Italy; antopal56@virgilio.it (A.P.); micheledebenedetto@hotmail.it (M.D.B.); 2Department of Rehabilitation, Cardiorespiratory Rehabilitation Unit, “V.Fazzi” Hospital, 73100 ASL Lecce, Italy; toraldodomenico@gmail.com; 3ENT & Oral Surgery Unit, Private Hospitals, 47121 Forli, Italy; filippomontevecchi72@gmail.com; 4Laboratory of Interdisciplinary Research Applied to Medicine (DReAM), University of Salento and ASL (Local Health Authority) at the “V Fazzi” Hospital, 73100 Lecce, Italy; luana.conte@unisalento.it; 5Laboratory of Advanced Data Analysis for Medicine (ADAM), Department of Mathematics and Physics “E. De Giorgi”, University of Salento, 73100 Lecce, Italy; 6Independent Scholar, Pisanelli no 25, 73020 Castrì di Lecce (LE), Italy; l.galasso78@gmail.com; 7“V.Fazzi” Hospital, Anesthesia and Intensive Care Department, 73100 ASL Lecce, Italy; filippo.derosa@alice.it (F.D.R.); caterinalattante@alice.it (C.L.); 8Department of History, Society and Human Studies, University of Salento, 73100 Lecce, Italy; enrico.ciavolino@unisalento.it; 9General Medicine, Univerzita Pavla Jozefa Safarika, 04001 Kosiciach, Slovakia; cate.arigliani10@gmail.com; 10Otolaryngology Head and Neck Surgery, University Hospital of Ferrara, 44124 Cona FE, Italy; claudio@claudiovicini.com; 11Head and Neck Department, ENT & Oral surgery Unit, G.B. Morgagni-L. Pierantoni, Hospital of Forlì, 47121 Forlì, Italy

**Keywords:** obstructive sleep apnea (OSA), drug-induced sleep endoscopy (DISE), propofol, Experimental 5 Video Stream System (5VsEs), bispectral index (BIS)

## Abstract

To illustrate a new technological advance in the standard drug-induced sleep endoscopy (DISE) model, a new machine was used, the Experimental 5 Video Stream System (5VsEs), which is capable of simultaneously visualizing all the decisional parameters on a single monitor, and recording and storing them in a single uneditable video. The DISE procedure was performed on 48 obstructive sleep apnea (OSA) or snoring patients. The parameters simultaneously recorded on a single monitor are (1) the pharmacokinetics and pharmacodynamics of propofol (through the target controlled infusion (TCI) pump monitor), (2) the endoscopic upper airway view, (3) the polygraphic pattern, and (4) the level of sedation (through the bispectral index (BIS) value). In parallel to the BIS recording, the middle latency auditory evoked potential (MLAEP) was also recorded and provided. Recorded videos from the 5VsEs machine were re-evaluated six months later by the same clinician and a second clinician to evaluate the concordance of the therapeutic indications between the two. After the six-month period, the same operator confirmed all their clinical decisions for 45 out of 48 videos. Three videos were no longer evaluable for technical reasons, so were excluded from further analysis. The comparison between the two operators showed a complete adherence in 98% of cases. The 5VsEs machine provides a multiparametric evaluation setting, defined as an “all in one glance” strategy, which allows a faster and more effective interpretation of all the simultaneous parameters during the DISE procedure, improving the diagnostic accuracy, and providing a more accurate post-analysis, as well as legal and research advantages.

## 1. Introduction

Obstructive sleep apnea (OSA) syndrome is a condition characterized by the presence of the complete or partial collapse of the upper airway during sleep (apnea and hypopnea, respectively). The consequences are sleep fragmentation associated with rapid intermittent hypoxia (IH) episodes with activation of the sympathetic nervous system and oxidative stress. In addition to the frequent presence of daytime sleepiness [1,2,3,4], OSA causes a wide spectrum of cardiovascular, metabolic, and neurocognitive comorbidities, more frequently associated with obesity [5]. In the last few years, studies on OSA have increased considerably, but in clinical practice, the disease is still highly underdiagnosed [6].

In OSA patients who need a better definition of the pathology, in surgical failures, or when patients do not accept continuous positive airway pressure (CPAP) therapy, an analytical procedure known as drug-induced sleep endoscopy (DISE) is used to better design the most suitable alternative treatment. This procedure allows the observation of the characteristics of the different levels of the upper airway, where soft tissue vibrations and/or an obstruction caused by collapse site(s) may be observed. It also allows the better definition of the functional alterations that cause poor adherence during CPAP treatment [7,8,9,10,11,12]. Notably, in this study, we performed DISE using propofol (propofol-DISE), since it allows for quickly obtaining an optimal level of sedation followed by a rapid post-sedation recovery [13,14,15,16,17]. This drug requires anesthesiologist management, and the use of a target-controlled infusion (TCI) pump is highly recommended [13]. During propofol-DISE, several parameters are evaluated simultaneously: endoscopic pattern; the pharmacodynamic and pharmacokinetic parameters through a TCI pump, which allows an optimal drug infusion modality and the continuous evaluation of drug concentration levels in the blood and brain [15,18,19,20,21,22]; polygraph recording synchronized with endoscopic images [23,24]; as well as the bispectral index (BIS) [25,26,27], i.e., the level of sedation achieved.

In the literature, several studies have tried to improve the DISE approach currently in use [9], whose major limitation is the subjectivity of the diagnostic–therapeutic decision, which may be associated with the lack of the recording of some parameters. At the moment, the parameters considered during the procedure are represented on several monitors and, with the exception of video endoscopy and polygraphic recording, the current technology does not provide for the recording and storage of the other parameters. In particular, the failure to record and store the pharmacokinetic and pharmacodynamic profiles, as well as the BIS values, that determine the endoscopic video pattern, does not allow an objective re-evaluation of the diagnosis, thus excluding the possibility of post-analysis [23,24,27,28,29].

The aim of this study was to propose a technological advance over the standard propofol-DISE model using a new machine called the 5 Video Stream Experimental System (5VsEs), which allows the simultaneous display of all the decisional parameters synchronized on a single monitor. The single video is recorded so that the video file cannot be modified or edited and is therefore suitable for use in research.

## 2. Materials and Methods

The DISE procedure was performed in accordance with the European Position Paper [14]. The following medical devices are part of the 5VsEs prototype: (1) a flexible endoscope (rhinolaryngoscope, 11101 series, Karl Storz^®^ CDD, Tuttlingen, Germany); (2) a compatible camera (image 1 222010 20, Karl Storz^®^ Tuttlingen, Germany); (3) an American Academy of Sleep Medicine (AASM)-compliant home sleep apnea testing device (Embletta Gold Portable Testing Device^®^, RemLogicE^®^ Software 2015, Embla System Inc, Broomfield, US) and its nasal canula and/or thermistors; (4) an oximeter (Nonin XPOD^®^, Plymouth, UK) with finger probe; (5) a BIS system (Covidien Ireland Limited^®^, Dublin, Irland); and (6) a TCI pump (Alaris PK^®^ by Carefusion PK, Basingstoke, UK), plus a middle latency auditory evoked potential (MLAEP) system (A-line^®^ sw version 1.5, owned by Danmeter A/S Denmark, Odense, Danmark). Detailed description of the 5VsEs is shown in Figure 1.

Patients provided their informed consent prior to participating in the study and international ethical standards were respected.

Age, sex, body mass index (BMI; mean 28, SD 3.6), Epworth Sleepiness Scale (ESS) (mean 12, SD 2.6), presence of comorbidities (38 patients (79%) with comorbidities and 10 (21%) without comorbidities), previous treatment (43 (90%) with no previous treatments and 5 (10%) with confirmed previous treatments), overnight polygraphic values of the apnea hypopnea index (AHI; mean 37 h/sleep (h/s), SD 17.1), oxygen desaturation index (ODI; mean 42 (h/s), SD 17.5), and lowest SatO_2_% (LOS; mean 77%, SD 11.1), defined according to Toraldo et al. [30] and AASM [8], are shown in Table 1. The sample was strictly random, so the number of women was random too.

Figure 2A shows the current DISE setting. Figure 2B shows the setting of DISE-polygraphy [24] in the first attempt to create a custom version. Figure 2C shows our propofol-DISE procedure set up prior to the use of 5VsEs, in which the operator controls five different data sources from five different devices. Figure 2D shows the operating room and the latest version of 5VsEs.

Figure 3 shows the final output of the 5VsEs represented on a single screen; all decisional parameters with optimized synchronization are displayed on a single high-definition (HD) monitor that integrates (i) the pharmacokinetics and pharmacodynamics of propofol (through the TCI pump monitor), (ii) the endoscopic upper airway view, (iii) the polygraphic pattern, and (iv) the level of sedation (through the BIS value). In parallel to the recording of the BIS, the middle latency auditory evoked potential (MLAEP) was also recorded and provided. The MLAEP system is already used in anesthesiology for the measurement of coma depth [31]; it was included for the first time in the DISE procedure. MLAEP was tested for a comparison with the BIS to evaluate which of the two is faster in determining the level of sedation. The results of this comparison are currently being validated.

The study was designed so that all procedures were performed by the same operator, who reported their clinical evaluations based on multiparametric observations on a 5VsEs single screen in the operation theater. The videos were re-evaluated 6 months later by the same operator (intra-operator and delayed evaluation for post-analysis) and by an external clinician (inter-operator evaluation) to evaluate the concordance of the therapeutic indications between the two. The second external operator also had access to the patient’s clinical anamnestic data. The second operator was blinded from the first operator’s impression so that the result was objective and impartial. The machine was validated by comparing the endoscopic video recording of standard DISE in parallel with 5VsEs. Figure 3 shows the final output of the 5VsEs.

### Ethics Approval

Research approval was obtained through the Ethics Committee of the Local Health Authority (ASL LE) at the Vito Fazzi Hospital (verbal no. 39, 26 November, 2019) and informed written consent was obtained from all research participants. Informed consent was obtained from all individual participants included in the study.

## 3. Results

A total of 48 OSA patients (43 men (90%) and five women (10%), mean age 49 years old, SD 12.5, range 18–75 years) were randomly enrolled in the study from February 2017 to November 2018 at the ENT Department at Vito Fazzi Hospital, ASL Lecce (Italy) (Table 1). Patients were recruited according to the following criteria: patients affected by snoring and/or OSA with surgery or mandibular advancement device (MAD) indication; patients affected by OSA with surgical failures and not compliant with CPAP treatment. Patients with anesthesia risk (ASA) ≤ 3, patients with a BMI ≤ 35, patients under 18 years old, pregnant women, and patients with contraindications to propofol infusion were excluded from the study.

After the six-month period (intra-operator delayed evaluation for post-analysis), the same operator confirmed all their clinical decisions for 45 out of 48 videos. Three videos (nos. 8, 10, and 22) were no longer evaluable due to recording failure, so were excluded from further analysis. The therapeutic decisions between the two different operators completely agreed in 91% of cases (41 cases out of 45), which is 98% considering that in cases 2, 16, and 44, the difference was only temporal, since the second operator, in contrast to the first, suggested a monitoring period before confirming the treatment or vice versa. In case no. 39, a low impact difference was found between both decisions. The last video out of 45 (case no. 12) was judged not evaluable by the second operator only, in accordance with normal medical judgments. Comparing the decisions made inside and outside the operating room (live/on monitor decision), data analysis seemed to confirm that both the operators were able to make a final clinical decision based on the recorded data in 44 cases out of a total of 45 (98%). The results are reported in Table 2.

## 4. Discussion

With the current standard DISE setting, some important details and information may be missed because the operator has to control several monitors simultaneously. The inability to save all parameters, with the exception of the video endoscopy and polygraphy [23,24,27,28,29], does not allow the subsequent re-evaluation of the entire procedure, excluding the possibility of a post-analysis and a second clinical opinion. The DISE procedure, allowing an endoscopic representation of the pathophysiology of OSA in a pharmacologically induced sleep situation, cannot be considered a simple endoscopy video; rather, it is a fluid concatenation of drug dynamic events linked to pathophysiological events that generate the endoscopic image moment by moment. Consequently, no correct post-analysis is possible when only the endoscopic and polygraphic pattern is stored without storing the drug dynamic context and the relative sedation level. The lack of pharmacokinetic and pharmacodynamics profiles storage in the current DISE makes the procedure highly subjective because the DISE procedure is partially documented. Instead, 5VsEs easily visualizes and stores the pharmacology of propofol through the TCI pump, which makes the procedure objective because it is documented and therefore verifiable at the time. Since the second operator was blinded from the first operator’s impression, the final decision was only based on the recorded data; the agreement/disagreement rate suggested the possibility of making diagnostic and therapeutic decisions based on the video and medical record documentation. In only one case, the agreement between the two operators was influenced by the variability of medical judgements. Under normal conditions, data recorded during the DISE procedure are displayed quickly on several monitors. When the manual transcription of data is required for clinical and study purposes, it may be inaccurate and often unreliable due to errors caused by rapid subsequent changes in values. The 5VsEs technology, conversely, allows complete data recovery through post-analysis.

In the literature [9], several research groups have highlighted the difficulty of standardizing results obtained during the standard DISE procedure, given the associated technical problems. In this paper, we illustrate the 5VsEs machine, which is a new approach and a technological advancement developed to overcome the issues with the standard procedure. The first target achieved was to display the five different signals for all the parameters, normally dissociated from each other, on a single monitor (video endoscopy, polygraphy, TCI, BIS, and MLAEP). The consequent effect was to construct an “all in one glance” approach enabled by the visualization and synchronization of the decisional parameters of DISE on a single monitor, guaranteeing a better perception that contributes to a more correct definition of the observation window. The optimized synchronization of all the monitors and the reduction and homogenization of the latency response between the different instruments allow the correct interpretation of the decision parameters and do not to interfere with the operator’s manual skills. The 5VsEs machine also ensures a certain speed and comfort of use, since the operator only has to place the sensors on the patient to start the procedure. In addition, to improve the quality of the endoscopic image and to better highlight the readability of the numerical values present in the polygraph traces, we also used HD video, which is rarely used in DISE [24]. Another important difference compared to the standard DISE is the possibility of recording the entire procedure, saving the output of the machine (Figure 3) on a removable electronic support, which can be used to re-evaluate the entire procedure at a later time, enabling better diagnostic management and providing information that can be used for educational and research aspects, including the possibility of using DISE in multicentric clinical trials and telemedicine. The software used produces a video file that cannot be modified, which enhances its scientific and medical–legal value. The recording of all the parameters deriving from the single screen also allows the storage of all the numerical data, not just the video; therefore, the intra-procedure numerical values (i.e., during the DISE), for example, of the SatO_2_% present in the video recording, can be easily checked and possibly recovered in the event of loss.

The intra-/inter-examination analysis and agreement between both clinicians is related to the “all in one glance” strategy, which depicts of the 5VsEs parameters on a single screen. The completely documented results of this analysis allow the operator (not present in the room) to see all the decision parameters so far not recorded and archived. The evaluation of the recording (Table 3) by the same operator after six months (intra-operator evaluation) provided information about the performance of the 5VsEs machine in the post-analysis phase in terms of our scientific research objective. The inter-operator evaluation provided information about multicentric study performance, medical–legal documentation, and education and teaching aspects.

These features could facilitate multicenter studies toward the standardization of the DISE procedure, which, to date, is performed in many different ways. In a systematic review [9], 17 studies were identified that proposed 14 new systems and three modified DISE classification systems to analyze anatomical results based on drug-induced sleep endoscopy. Inter-observer agreement between an experienced observer and an observer in training proposed by Carrasco-Llatas et al. [32] can be easily achieved through the 5VsEs machine. The machine’s advantages over the standard DISE are listed in Table 3.

### Study Limitations

Three videos were excluded was due to recording failure. The recordings were sometimes interrupted or discontinued due to the insufficiency of the sensors or when the registration procedure was interrupted for live optimization and live development of the 5VsEs prototype.

## 5. Conclusions

The 5VsEs machine allows a continuous and optimized evaluation and storage of all useful parameters due to the “all in one glance” approach made possible by the integration and synchronization of all DISE parameters on a single monitor. Unlike the standard procedure, 5VsEs permits a non-modifiable full multiparameter recording. The 5VsEs machine was conceived to solve some issues with the method, including methods of the classification of endoscopic patterns, drug infusion techniques, etc. With the 5VsEs prototype, the operator can continuously evaluate the drug kinetic and pharmacodynamic profiles during the procedure and during re-evaluation in post-analysis and research, which are indispensable for correct evaluation in the post-analysis of the endoscopic pattern along with the BIS data and polygraphic findings. In some situations, the recording was discontinuous, so the next effort will focus on optimizing the prototype. However, this new prototype represents a technological advance in the DISE procedure, allowing for better perception of the observation window. This new diagnostic model needs further studies to validate its reliability in clinical practice on a larger number of patients. The main goal of this work was to present a prototype with interesting and promising potential to improve the management of the DISE procedure and research.

## Figures and Tables

**Figure 1 ijerph-17-04261-f001:**
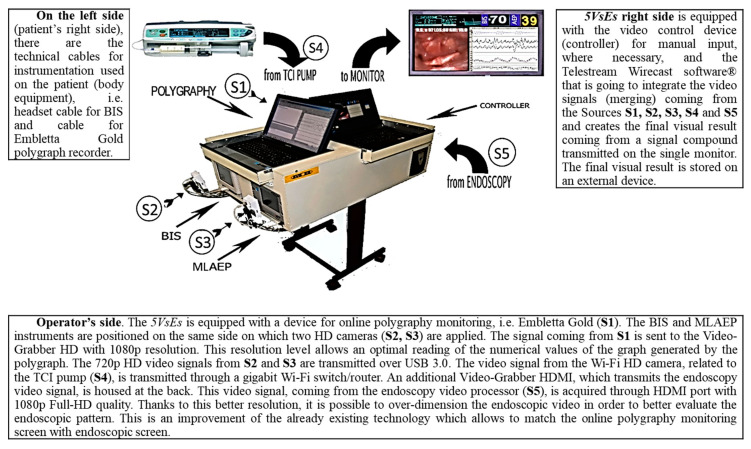
Detailed description of the Experimental 5 Video Stream System (5VsEs).

**Figure 2 ijerph-17-04261-f002:**
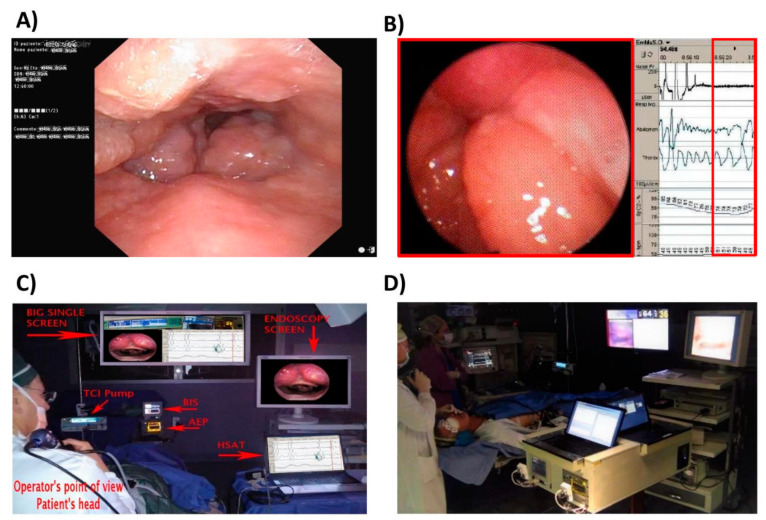
(**A**) Current drug-induced sleep endoscopy (DISE) setting; (**B**) customized DISE-polygraphy; (**C**) propofol-DISE. Prior to the use of the 5VsEs, the operator managed five different data sources from five different devices. (**D**) The operating room and the 5VsEs.

**Figure 3 ijerph-17-04261-f003:**
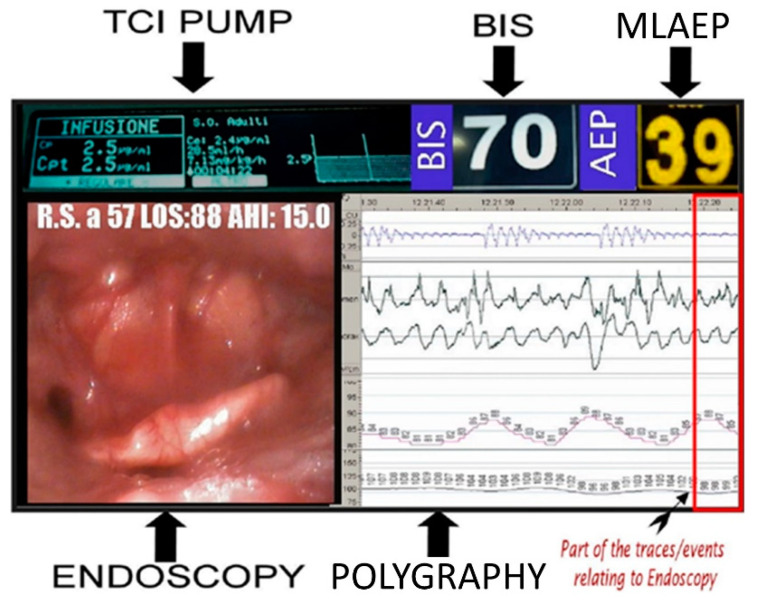
The 5VsEs single screen (“all in one glance” strategy).

**Table 1 ijerph-17-04261-t001:** Patient characteristics.

ID Number	M/F	BMI	ESS	Comorbidities	Previous Treatment	Age (years)	Drug-induced Sleep Endoscopy (DISE) Date	AHI (h/s)	ODI (h/s)	LOS (%)
**1**	M	28	15	Y	N	48	February 2017	26	25	80
**2**	M	30	14	Y	N	43	February 2017	18	31	81
**3**	M	31	18	Y	N	52	February 2017	56	50	68
**4**	M	32	13	Y	N	52	February 2017	65	77	62
**5**	M	33	12	Y	Y	45	April 2017	12	22	94
**6**	M	34	9	N	N	64	April 2017	26	30	80
**7**	M	24	10	N	N	58	April 2017	27	32	88
**8**	M	29	16	Y	N	53	April 2017	43	46	64
**9**	F	34	15	Y	N	58	April 2017	53	57	57
**10**	M	31	14	Y	N	56	April 2017	57	49	84
**11**	M	28	12	Y	N	75	May 2017	43	49	73
**12**	M	23	11	Y	N	71	May 2017	43	47	78
**13**	M	34	10	Y	N	54	May 2017	56	73	55
**14**	M	27	15	N	N	41	May 2017	22	18	82
**15**	M	28	10	Y	N	47	June 2017	71	82	78
**16**	F	28	13	Y	Y	55	June 2017	32	36	64
**17**	F	19	10	N	N	41	June 2017	12	22	94
**18**	M	25	13	Y	N	28	July 2017	27	29	92
**19**	M	27	14	Y	N	45	July 2017	24	28	89
**20**	M	35	18	Y	N	41	July 2017	74	79	59
**21**	M	25	13	Y	N	28	September 2017	27	29	92
**22**	M	31	11	Y	N	46	September 2017	58	55	70
**23**	M	25	13	Y	N	48	September 2017	26	33	82
**24**	M	25	10	N	N	19	September 2017	27	38	80
**25**	M	31	11	Y	N	41	November 2017	30	35	84
**26**	M	27	12	Y	N	58	November 2017	25	35	83
**27**	M	28	12	Y	N	40	November 2017	43	48	72
**28**	M	26	11	N	N	60	November 2017	22	18	88
**29**	M	27	12	N	N	38	November 2017	28	27	78
**30**	M	33	14	Y	N	53	January 2018	36	36	72
**31**	M	32	10	Y	N	51	January 2018	50	51	77
**32**	M	26	12	Y	N	56	February 2018	25	37	81
**33**	M	23	14	Y	N	55	February 2018	23	30	89
**34**	M	27	12	Y	N	54	February 2018	52	55	69
**35**	M	34	11	Y	N	68	March 2018	47	54	63
**36**	M	25	13	Y	N	29	May 2018	27	32	82
**37**	M	23	9	Y	N	35	May 2018	26	26	84
**38**	M	29	11	Y	N	18	May 2018	58	56	53
**39**	M	33	6	N	N	57	June 2018	27	33	78
**40**	M	28	9	N	N	62	June 2018	27	31	83
**41**	M	28	14	Y	Y	60	June 2018	22	29	88
**42**	F	31	13	Y	Y	56	July 2018	78	75	59
**43**	F	29	19	Y	N	28	October 2018	57	61	62
**44**	M	23	10	Y	N	57	October 2018	20	27	88
**45**	M	25	10	Y	N	46	October 2018	35	72	70
**46**	M	27	12	Y	N	55	October 2018	25	32	87
**47**	M	30	12	N	N	55	November 2018	55	57	79
**48**	M	27	6	Y	Y	36	November 2018	15	18	90

ID: identification number; M/F: male/female; BMI: body mass index; ESS: Epworth Sleepiness Scale; AHI (h/s): apnea hypopnea index; ODI (h/s): oxygen desaturation index; and LOS (%): lowest SatO_2_%.

**Table 2 ijerph-17-04261-t002:** Results of DISE interpretations based on 5VsEs in the operating room and after 6 months by the same operator compared to a second external operator.

ID Number	Instant Decision	Same Operator (6 Months After)	Second Operator
**1**	BRP	SAME	SAME
**2**	BRP	SAME	SAME after monitoring
**3**	BRP	SAME	SAME
**4**	BRP	SAME	SAME
**5**	No treatment	SAME	SAME
**6**	FEP	SAME	SAME
**7**	BRP	SAME	SAME
**8**	BRP	Unjudgeable Video Recording	SAME
**9**	PSG Lab	SAME	SAME
**10**	BRP	Unjudgeable Video Recording	SAME
**11**	BRP	SAME	SAME
**12**	BRP + Epiglottoplasty	SAME	Unjudgeable Video Recording
**13**	CPAP	SAME	SAME
**14**	FEP + MAD	SAME	SAME
**15**	MMA	SAME	SAME
**16**	BRP + MAD	SAME	BRP Monitoring before MAD
**17**	MAD	SAME	SAME
**18**	BRP	SAME	SAME
**19**	BRP + MAD	SAME	SAME
**20**	BRP	SAME	SAME
**21**	BRP + MAD	SAME	SAME
**22**	BRP + TORS	Unjudgeable Video Recording	SAME
**23**	BRP + Thyro-Hioido-Pexy	SAME	SAME
**24**	BRP	SAME	SAME
**25**	FEP	SAME	SAME
**26**	FEP	SAME	SAME
**27**	BRP + GlossoEpiglottopexy	SAME	SAME
**28**	BRP	SAME	SAME
**29**	BRP	SAME	SAME
**30**	Septoplasty + BRP	SAME	SAME
**31**	BRP	SAME	SAME
**32**	BRP	SAME	SAME
**33**	No treatment	SAME	SAME
**34**	BRP	SAME	SAME
**35**	BRP and Monitoring	SAME	SAME
**36**	BRP and Monitoring	SAME	SAME
**37**	Septoplasty + BRP	SAME	SAME
**38**	MMA + Epiglottoplasty	SAME	SAME
**39**	BRP + MAD	SAME	BRP
**40**	FEP	SAME	SAME
**41**	BRP + MAD	SAME	SAME
**42**	BRP + MAD	SAME	SAME
**43**	diet + CPAP	SAME	SAME
**44**	Wait & See for Epiglottoplasty	SAME	Epiglottoplasty and monitoring
**45**	BRP	SAME	SAME
**46**	BRP	SAME	SAME
**47**	BRP and Monitoring	SAME	SAME
**48**	MAD	SAME	SAME

FEP: Functional expansion pharyngoplasty; PSG-lab: polysomnography in laboratory; BRP: barbed reposition pharyngoplasty; MAD: mandibular advancement device; MMA: maxillo-mandibular advancement; TORS: trans-oral robotic surgery; CPAP: continuous positive airway pressure.

**Table 3 ijerph-17-04261-t003:** 5VsEs vs. standard technology performance.

5VsEs vs. Standard Procedure	5VsEs	Standard
**Documentation of pharmacokinetics–pharmacodynamics profiles**	YES	NO
**Documentation of the observation window**	YES	NO
**Documentation useful for multicentric studies**	YES	NO
**Recovery of lost data during data collection, useful for research purposes**	YES	NO
**Statistical analysis thanks to data collection confidence (uneditable video documentation)**	YES	NO
**Medical–legal documentation more complete**	YES	NO
**Improvement of education and teaching aspects**	YES	NO
**Efficient telemedicine**	YES	NO

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
