# Peer review of "A New Technological Advancement of the Drug-Induced Sleep Endoscopy (DISE) Procedure: The “All in One Glance” Strategy"

_ijerph, 2020, doi:10.3390/ijerph17124261_

Round 1
Reviewer 1 Report
Innovative approach to DISE that could markedly improve diagnostics in OSA. I would try to refine the abstract including which 5 parameters are included in the "one-glance screen".
Acceptable but not customary to insert the studied population into the methods section and not into the results section.
Author Response
Point 1. Innovative approach to DISE that could markedly improve diagnostics in OSA. I would try to refine the abstract including which 5 parameters are included in the "one-glance screen".
Response 1. Ok, we refined the abstract including the 5 parameters. (please, see line 32)
Point 2. Acceptable but not customary to insert the studied population into the methods section and not into the results section.
Response 2. Please, refers to line 164. We inserted the studied population into the results section.
Point 3. Insert the limitation of the study into discussion
Response 3. Please, refers to line 265. We inserted the limitation of the study into discussion.
Point 4. The results take on too much space into abstract.
Response 4. We deleted part of the results into abstract. Please see line 41.
Reviewer 2 Report
Overall, the article is a successful piece of work and enriches science. A very good achievement! Thank you for this valuable work.
Before I come to individual comments, I have to add one thing: I do not understand the intra / interrater examination in the study. You describe the new system very nicely (it is an interesting innovation) with all the advantages. What does the agreement of N = 2 tell us with 2 examiners? They describe the agreement in the results section, but do not go into it in the discussion. In my view, it is not possible to derive this analysis for the "All at a Glance" strategy.
In my view, this should definitely be supplemented in the discussion with an indication that this result does not allow any statement about the objectivity of the system. The two examiners would therefore only have been chosen as an example for a first presentation of the 5VsEs machine in order to demonstrate an example of an advantage of the system.
Line 90: Children were also excluded because your age was given between 18-75 years?
Line 104: you mean the Epworth sleepiness scale (ESS)?
Line 108: I would add the units (for example: per hour in AHI), the same in ODI
Line 112: The ESS appears again incorrectly described in the legend of the table.
Line 116: What means HSAT in Figure I?
Line 175: The lack of pharmacokinetic and pharmacodynamics profiles storage makes the procedure highly subjective à When reading this sentence, you get the impression that working with the 5VsEs machine objectifies the whole. It is a comment similar to the one mentioned above. In my view, the proof is not provided with N = 2. For this, a validation (which you also describe at the end of the work) must be carried out first.
Line 197 and 210 (Table 3)
“Recovery of lost data during data collection, useful for research purposes.” This Point, I don’t understand it. We also use a Storz technology system with an external memory stick. I save all endoscopic videos there. Do you mean saving all data?
Author Response
Point 1. Line 90: Children were also excluded because your age was given between 18-75 years?
Response 1. Right. We now have reported age under 18 exclusion on line 170.
Point 2. Line 104: you mean the Epworth sleepiness scale (ESS)?
Response 2. Yes, thank you. We have now corrected the acronymous. Please see line 109 and 174.
Point 3. Line 108: I would add the units (for example: per hour in AHI), the same in ODI
Response 3: OK, thanks. We have now inserted the units. Please, see line 113.
Point 4. Line 112: The ESS appears again incorrectly described in the legend of the table.
Response 4. That’s true. We have now corrected in the legend of the table.
Point 5. Line 116: What means HSAT in Figure I? Sorry, it should be an old misprint.
Response 5. We have now substitute HSAT with Polygraphy. Please, see figure 1.
Point 6. Line 175: The lack of pharmacokinetic and pharmacodynamics profiles storage makes the procedure highly subjective à When reading this sentence, you get the impression that working with the 5VsEs machine objectifies the whole. It is a comment similar to the one mentioned above. In my view, the proof is not provided with N = 2. For this, a validation (which you also describe at the end of the work) must be carried out first.
Response 6. In the current DISE, the lack of archiving of pharmacokinetic and pharmacodynamic profiles makes the procedure highly subjective because it is not fully documented. The instrument we propose, instead, allows the archiving of pharmacological parameters making them finally documentable and verifiable at a later time. The person who is not present in the operating room, therefore, can look again at the data collected and express his or her point of view. Please, refer to line 247 and in line 150.
Point 7. Line 197 and 210 (Table 3): “Recovery of lost data during data collection, useful for research purposes.” This Point, I don’t understand it. We also use a Storz technology system with an external memory stick. I save all endoscopic videos there. Do you mean saving all data?
Response 7. The recording of all the parameters deriving from the single screen allows to store also all the numerical data, not only the video data. Those intraprocedural numerical values (i.e. during the DISE) e.g. the O2 saturation can be easily checked and eventually recovered. Please refer to line 242.
Reviewer 3 Report
In this article, the authors are trying to introduce a new technological advancement in the arena of Drug-Induced Sleep Endoscopy (DISE). As we all know, DISE is usually performed in the operating room with patients receiving propofol and monitored with multiple parameters making it difficult to use as an imaging modality in extensive clinical studies. I feel this innovative technology would address that issue in the future.
I have some recommendations:
- I see women are underrepresented in the study population, but an explanation addressing this issue would be reasonable. As the study is comparing technology, I don't think that sex/gender would have a significant impact. But it is worth to mention it somewhere.
- It was not mentioned whether the second operator was blinded from the first operator's impression.
- Understandably, there might be some inter-observer variability, which is typical in the clinical decision between the physicians. I would be more interested in whether the second operator could come up with the final decision based on the recorded data.
- I would recommend including an analysis of whether both the operators were able to come up with a final clinical decision based on the recorded data after 6 months. I think this would be 45 vs. 44 cases.
- It is better to explain why three videos were excluded from the analysis(more detail than the technical reason) and why the second operator in detail did not judge 45th video.
- The conclusion can be shorter. Authors don't have to mention aim again in the conclusion session along with other advantages of recorded videos like "diagnostic, educational, research and medical-legal purposes at a later time." It was already reported in the discussion.
Author Response
Point 1. I see women are underrepresented in the study population, but an explanation addressing this issue would be reasonable. As the study is comparing technology, I don't think that sex/gender would have a significant impact. But it is worth to mention it somewhere.
Response 1. The sample was strictly random so the number of women was random too. Please, see line 116.
Point 2. It was not mentioned whether the second operator was blinded from the first operator's impression.
Response 2. The second operator was blinded from the first operator's impression so that the result was objective and impartial. We have now clarified it on line 149.
Point 3. Understandably, there might be some inter-observer variability, which is typical in the clinical decision between the physicians. I would be more interested in whether the second operator could come up with the final decision based on the recorded data.
Response 3. Since the second operator was blinded from the first operator's impression, the final decision is only based on the recorded data: the agreement between the two clinicians suggests the possibility of making a diagnostic and therapeutic decision based only on the documentation obtained from the 5VsES machine and the patient's medical records.
Point 4. I would recommend including an analysis of whether both the operators were able to come up with a final clinical decision based on the recorded data after 6 months. I think this would be 45 vs. 44 cases.
Response 4. Operators were able to come up with a final clinical decision based on the recorded data in 44 cases on a total of 45 (98%). Please see line 188 where we clarified it much better.
Point 5. It is better to explain why three videos were excluded from the analysis (more detail than the technical reason) and why the second operator in detail did not judge 45th video.
Response 5. Three videos of the excluded cases was due to insufficient recording. The recording can sometimes be interrupted discontinuous for insufficiency of the sensors or when the registration procedure had to be interrupted for the optimization and the direct development of the 5VsEs prototype. Only the second operator does not judge case n. 12 and this is in accordance with the normal variability of medical judgements. Please refer to line 187 and 216.
Point 6. The conclusion can be shorter. Authors don't have to mention aim again in the conclusion session along with other advantages of recorded videos like "diagnostic, educational, research and medical-legal purposes at a later time." It was already reported in the discussion.
Response 6. Ok, thanks, we tried to improve the conclusions as well. Please see Conclusions.
Round 2
Reviewer 2 Report
Thank you for the explanations, changes and additions.
The article reads very nicely.
I personally still find it critical to compare therapy proposals. It is known that the "opinions" within a DISE fluctuate a lot (regarding the therapeutic possibilities). Even though it is a significant advantage to have the same video viewed and rated by several specialists, I am surprised that the agreement is so high.
I would be interested in the agreement of treatment proposal when 10 evaluators see the same DISE-Video with all information’s. That might be an idea for the future. We often say: “3 doctors, 3 different treatments”.
A small change would still have to be made. As far as I know, the ODI is also given per hour (not%). (based on the AHI)
Thank you for your nice overview. Overall, I see in your article an enrichment of science in the planning of surgical measures in patients OSAS.
Author Response
Thank you for your helpful notes to this manuscript. We agree with your last comment. We must also say that the comparison between the two operators was made by choosing them from the same department, so that they have a similar expertise and background extraction. Do you think we should write this in the manuscript at some point? many thanks in advance